

# **Technical note: Accelerate coccolith size separation via repeated centrifugation**

Hongrui Zhang[1,2], Chuanlian Liu[1], Luz Maria Mejia[2], Heather Stoll[2]

1. State Key Laboratory of Marine Geology, Tongji University, Shanghai, 200092, China

2. Geological Institute, Department of Earth Science, Sonneggstrasse 5, ETH, 8092, Zürich, Switzerland

Corresponding to: Hongrui Zhang (103443_rui@tongji.edu.cn; zhh@ethz.ch)

## **Abstract:**

Coccolithophore play a key role in the marine carbon cycle and ecosystem. The carbonate shells produced by coccolithophore, named as coccolith, could be well preserved in the marine sediment for million years and become an excellent archive for paleoclimate studies. The micro filtering and sinking-decanting method have been successfully designed for coccolith separation and promoted the development of geochemistry studies on coccolith, such as the stable isotopes and Sr/Ca ratio. However, these two methods are still not efficient enough for the sample-consuming methods. In this study, the trajectory of coccoliths movement during a centrifugation process was calculated in theory and carefully tested by separations in practice. We offer a matlab code to estimate the appropriate parameter, angular velocity at a fixed centrifugation duration, for separating certain coccolith size fractions from bulk sediment. This work could improve the efficiency of coccolith separation, especially for the finest size fraction and make it possible to carry the clumped isotope and radio carbon analysis on coccolith in sediment.



## 1. Introduction

Coccolithophores are a group of marine calcifying eukaryotic phytoplankton., whose calcite exoskeletons (i.e. coccolith) contribute significantly to the particulate inorganic carbon (PIC) export from the euphotic zone into the deep ocean (Young and Ziveri, 2000). Coccoliths preserved in marine sediment are also excellent archive for paleo-productivity reconstruction (Beaufort et al., 1997). The element ratio, Sr/Ca, in coccolith is correlated with the growth rate of calcite crystal (Stoll et al., 2002) thereby becoming a competitive candidate for coccolithophore growth rate which is an essential parameter in the paleo-$CO_2$ reconstruction by alkenone carbon isotope. However, the coccolith geochemical analyses are limited by the difficulty of separating coccolith from bulk sediment. To solve this problem, different separating methods have been proposed in the past a few decades (Paull and Thierstein, 1987; Stoll and Ziveri, 2002; Minoletti et al., 2008).

Most of them, in general, could be categorized into two groups: the first one is micro-filtering and the second is sinking-decanting technique. The micro-filtering method relies heavily on the specifications of micro filter membrane (such as 3μm, 5μm and 8μm pore size), which is highly effective in separation of the larger size coccoliths, but useless for coccolith smaller than 2μm. The sinking-decanting method, on the other hand, could offer more freedom in coccolith size by adjusting the sinking durations, thereby separating both small and large coccoliths. However, because of the slow sinking speed, a single separation of particles smaller than 2 μm may take more than 10 hours in settling. Moreover, about 6–8 times operations should be repeated, which means a full separation may takes at most one week. Hence, it is necessary to improve this method by reducing the time cost in coccolith separation.

Based on the Stokes sinking equation, the sinking rate of a certain particle increases with the increase of density difference between particle and liquid, decrease of the liquid viscosity and the increase of gravity. Changing the physical property of liquid often leads to the organic and toxic solvent which could lead to potential





contaminations for the further geochemistry analyses. A better way to accelerate
coccolith sinking speed is changing the gravity, or the acceleration speed of the
reference system, which can be easily achieved by centrifugation. One study has
mentioned the usage of centrifugation in coccolith separation, but only centrifugation
settings for a special case were provided (Hermoso et al., 2015). Here in this study, the
method of separation coccolith by centrifugation is introduced systemically. We first
calculate the trajectory of coccolith movement in a centrifugation processes and show
how to estimate the centrifugation parameters in different situations. After that, two
tests are performed to confirm the robustness of our calculations. Ultimately, a sample
containing coccoliths ranging from 2 μm to 12 μm is selected for a separation case in
practice.
**2. Trajectory of coccoliths during centrifugation**

The movement of coccolith in centrifugation is similar to that under the gravity.

Previously, we have calculated the separation ratio variation with time during the
settling (Zhang et al., 2018). All calculations in this study are with an assumption that
the coccolith is in the force balance all the time during both settlings and centrifugations
for a convenience of calculation. Here we offer a brief proof for this assumption and do
a quick review of derivation we did before.

Based on the Newton second law, the force balance of a sphere object during

sinking can be described by the following equation:
$$F = \frac{4}{3}\pi r^3 \rho_p g \; - \; \frac{4}{3}\pi r^3 \rho_l g - 6\pi\eta rv = \frac{4}{3}\pi r^3 \rho_p \frac{dv}{dt} \qquad (\text{Eq. 1.})$$
where $F$ is the join force of particle, which is equal to zero in force balance, $r$ is
the radium of sphere, $\rho_p$ and $\rho_l$ are the density of particle and liquid, respectively, $\eta$ is
the velocity of liquid and $v$ is the particle sinking speed, $dv/dt$ is the particle acceleration
speed, which can be also marked as $a$. On the right side of the first equal mark, the first
term is the gravity force, the second term is buoyancy and the third term is the dragging





force from liquid. Transform Eq. 1, we can obtain the expression of accelerated speed
(a = F/m) of sphere as Eq. 2:
$$a = \frac{dv}{dt} = -\frac{9\eta}{2r^2}v + \frac{g}{\rho_{cal}}(\rho_p - \rho_l) \qquad (\text{Eq. 2.})$$

Given the initial value of sinking speed is equal with zero at the initial time (t = 0),
we can solve the differential equation Eq. 2 and obtain the variation of velocity with
time as following equation:
$$v = \frac{-e^{\left[-\frac{9\eta}{2r^2}t + \ln\left(-\frac{g}{\rho_{cal}}(\rho_p - \rho_l)\right)\right]} + \frac{g}{\rho_{cal}}(\rho_p - \rho_l)}{\frac{9\eta}{2r^2}} \qquad (\text{Eq. 3.})$$

when the value of t is large enough, the first term of numerator in Eq. 3 is close to
zero, which represents the sinking velocity is close to the termination sinking velocity
described in Stocks equation (Eq. 4).
$$\lim_{t \to \infty} v = \frac{2(\rho_p - \rho_l)gr^2}{9\eta} \qquad (\text{Eq. 4.})$$

Given the particle as a 5 μm in radium calcite carbonate sphere with a density of
2.7 g cm$^{-3}$ and the density of water is equal to 1.0 g cm$^{-3}$, when the t is equal to 10$^{-7}$ s,
the first term of numerator is 3.7×10$^{-44}$ m s$^{-2}$ and small enough to be neglected compared
with the second term, which is 6.3 m s$^{-2}$. The time scale in coccolith separation is minute
for centrifugation and hour for settling, therebefore we suggest that it is reasonable to
assume the coccolith sinks with the 'terminal speed' from the very beginning.
The only difference between the terminal speed in centrifugation and under gravity
is the acceleration speed. If the g in Eq. 1 – 4 is adapted by a, which is the acceleration
speed of coccolith during centrifugation, these four equations above can also describe
the sphere movement in the centrifugation if we adapt the gravity to centripetal
acceleration (ca). Here we define a new parameter named as Sinking Parameter (*sp*):
$$sp = \frac{v}{g} \qquad (\text{Eq. 5.})$$



The physical meaning of *sp* is the influence of coccolith shape and liquid property
(density and viscosity) on sinking velocity without considering the effect of gravity (or
the acceleration rate of reference system). The sinking speed of coccolith in water
during a centrifugation (v') can be described as following:
$$v' = sp \times ca = sp \times \omega^2 \times (L + D) \qquad (\text{Eq. 6.})$$
where the ca is centripetal acceleration during centrifugation, ω is angular velocity
of centrifuge, the (L+D) is the rotation radium as illustrated in **Figure 1**. The L is a
fixed value for a certain type of centrifuge and the D depended on the position of
coccolith in the tube. Here we should notice two issues: the first one is that the rotation
radius is varying when coccolith is moving in the centrifuge tube, in other words, D is
always changing. This effect could be ignored when the L is much larger than D, but,
unfortunately, most of centrifuge employed in geochemistry laboratory is not large
enough. The second one is the angular velocity is dynamic during when the centrifuge
is accelerating and decreasing. To solve these two dynamic parameters, Eq.6 was
transformed into a form of differential equation as Eq. 7 for the convenience of
integration in the next step.
$$dt = \frac{dD}{v} = \frac{dD}{sp \times \omega^2 \times (L + D)} \qquad (\text{Eq. 7.})$$
For all centrifugations there are three stages: the acceleration stage ($t_1$ to $t_2$ in
Figure 1), the constant angular velocity stage ($t_2$ to $t_3$ in Figure 1) and the deceleration
stage ($t_3$ to $t_4$ in Figure 1). The duration of acceleration stage and deceleration stage can
usually be controlled and the angular velocity is changing with a constant speed. For
those machines which the angular velocity dynamic ($\omega = f(t)$) is unknow we should
measure it manually.





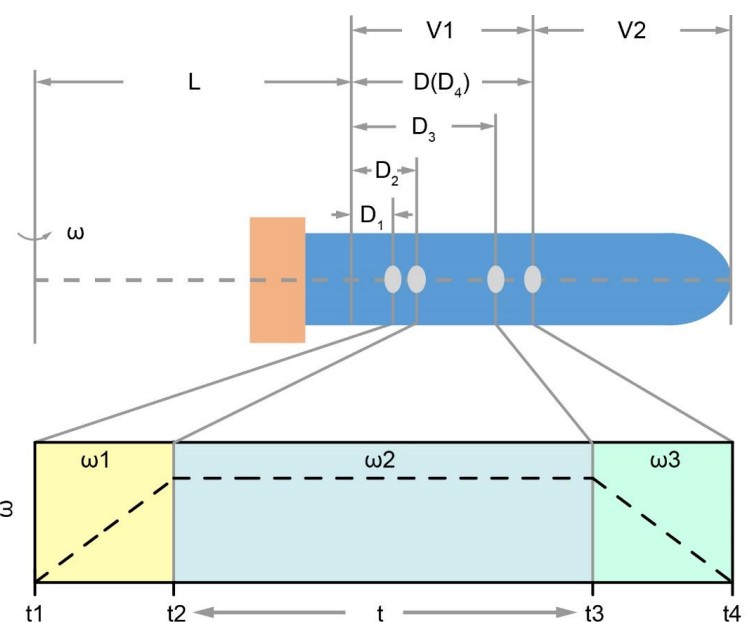


**Figure 1.** The position of coccolith and the variation of ω in the three centrifuging stages: L
represents the minimum rotation radium, the $V_1$ and $V_2$ represent the volume of two parts; in the
first stage, the angular velocity increases from zero to ω1 (it could be linear or cubic, which
depends on the machine). Meanwhile the coccolith moves a distance of $D_2$-$D_1$.; similarly, the
coccolith moves a distance of $D_3$-$D_2$ in the second stage and it march a distance of $D_4$-$D_3$ in the
last stage.
After knowing the angular velocity curve, integrate the D over t in the Eq. 7 by
three steps from $t_1$ to $t_4$:
$$sp \times \int_{t_1}^{t_2} \omega_1{}^2 dt = \ln^{(L+D_2)} - \ln^{(L+D_1)} \qquad (\text{Eq. } 8.)$$

$$sp \times \int_{t_2}^{t_3} \omega_2{}^2 dt = \ln^{(L+D_3)} - \ln^{(L+D_2)} \qquad (\text{Eq. } 9.)$$

$$sp \times \int_{t_3}^{t_4} \omega_3{}^2 dt = \ln^{(L+D_4)} - \ln^{(L+D_3)} \qquad (\text{Eq. } 10.)$$





Add the Eq. 8 – 10 together gives:

$$
sp \times \left( \int_{t_1}^{t_2} \omega_1{}^2 \mathrm{dt} + \int_{t_2}^{t_3} \omega_2{}^2 \mathrm{dt} + \int_{t_3}^{t_4} \omega_3{}^2 \mathrm{dt} \right) = ln^{(L+D_4)} - ln^{(L+D_1)} \qquad (Eq.\,11.)
$$

Set $D_4$ equal to D, which represents the maximum distance that a coccolith can
move in the upper suspension $V_1$. Now we can use the coccolith sinking property, *sp*,
and centrifugation settings to describe the coccolith position after centrifugation $D_1$:

$$
D_1 = \frac{L + D}{e^{\left[ sp \times \left( \int_{t_1}^{t_2} \omega_1{}^2 \mathrm{dt} + \int_{t_2}^{t_3} \omega_2{}^2 \mathrm{dt} + \int_{t_3}^{t_4} \omega_3{}^2 \mathrm{dt} \right) \right]}} - L \qquad (Eq.\,12.)
$$

The meaning of $D_1$ is all coccolith with an initial position on the right side of $D_1$
in **Figure 1** will move to the right side of $D_4$ and then be kept in the suspension after
pumping, while the coccolith on the left side of $D_1$ will be removed by pumping.
In our previous publication (Zhang et al., 2018), we defined a parameter named as
separation ratio (R), which represents the percentage of coccolith removed in one
separation if we pump the upper $V_1$ volume suspension out of ($V_1 + V_2$) suspension in
total.

$$
R = \frac{V_1 \times \dfrac{D_1}{D}}{V_1 + V_2} \qquad (Eq.\,13)
$$

Replacing the $D_1$ in Eq. 15 with Eq. 12 gives the separation ratio (R) as a function
of centrifugation settings:

$$
R = \frac{V_1}{V_1 + V_2} \times \frac{1}{D} \times \left( \frac{L + D}{e^{\left[ sp \times \left( \int_{t_1}^{t_2} \omega_1{}^2 \mathrm{dt} + \int_{t_2}^{t_3} \omega_2{}^2 \mathrm{dt} + \int_{t_3}^{t_4} \omega_3{}^2 \mathrm{dt} \right) \right]}} - L \right) \qquad (Eq.\,14)
$$

The R can be employed in estimating the centrifugation parameters for separating
one type of coccoliths from another. For example, if we want to separate a group of
coccolith (marked as CoccolithA, with sinking parameter $sp_A$) from another group of
coccolith (marked as CoccolithB, with sinking parameter of $sp_B$ and $sp_A < sp_B$), the R of
CoccolithB should be set as zero, which means all CoccolithB in the section $V_1$ have
sunk into $V_2$ after centrifugation and therefore all coccolith pumped out should be





Coccolith$_A$. To solve the angular velocity ($\omega_2$) and centrifugation duration (t = t$_3$-t$_2$) in
Eq.14, we need to fix at least one of them. Usually the duration could be safely set as 1
min or 2 min, then solve the suitable angular velocity with known parameters V$_1$, V$_2$,
D and L. The matlab code for the parameter estimation is in attachment. After repeating
these 'centrifugation-pumping' routines several times, the Coccolith$_A$ could be fully
separated from Coccolith$_B$.
**3.Test of the correctness of calculations**
**3.1 Experimental design**

To test the robustness of our estimation in the last section, we performed two

groups of experiments comparing the observed with predicted separation ratio. Here we
select two different coccoliths, *F. profunda* and small *Gephyrocapsa*, with small size
and thereby slow sinking speed sampled from ODP 807 and IODP U1304, respectively.
Most of small *Gephyrocapsa* employed in this study are smaller than 3 μm with a
mixture of *G. muellerae* less than 10%. Two centrifuges from Anting Company, TDL–
40B and DL–5B, were selected to perform the tests. The angular velocity of DL-5B can
be set as linear increased or decreased with time in the acceleration or deceleration
stages, while the angular velocity of TDL–40B was measured manually by reading the
number on the instrument panel. The centrifugation duration can only be adapted by a
step of one minute on both of these two machines. The slowest angular velocities of
these two machines are 500 revolutions per minute (rpm). If we selected the water as
dispersion agent, most of the coccolith we used will sink to the tube bottom after two
minutes even with the slowest angular velocity. Hence, to slow down the coccolith
sinking speed in these tests, glycerol solution was employed in this equation test, which
can be dissolved with water in any proportion and washed away from carbonate calcite
particles conveniently. The density and viscosity data can be found in **Table 1**.

All calculations above are for the situation that particles sinking in the water or the

diluted solution, the physical property of which is close to water. However, in this case,





the property of glycerol is significant different with water. Here we define a new
parameter, τ, to transform the sinking speed in water to that in different liquid. The
physical meaning of τ is a ratio turning the sinking velocity in water (v) to the velocity
in any liquid with different density and viscosity (v'):
$$v' = v \times \tau \qquad \text{(Eq. 15)}$$
Based on the definition of Stokes equation, the term τ can be calculated as
following:
$$\tau = \frac{(\rho_p - \rho_l)}{(\rho_p - \rho_w)} \times \frac{\eta_w}{\eta_l} \qquad \text{(Eq. 16)}$$
where the $\rho_p$, $\rho_l$ and $\rho_w$ are density of particle, liquid (in this study is glycerol
solution) and water; the $\eta_l$ and $\eta_w$ are the viscosity of liquid and water.
Combine the Eq. 14–16 forming the separation ratio as a function of centrifugation
settings in different liquid:
$$R = \frac{V_1}{V_1 + V_2} \times \frac{1}{D} \times \left( \frac{L + D}{e^{\left[ \frac{v}{g} \times \frac{(\rho_p - \rho_l)}{(\rho_p - \rho_w)} \times \frac{\eta_w}{\eta_l} \times \left( \int_{t_1}^{t_2} \omega_1{}^2 \mathrm{dt} + \int_{t_2}^{t_3} \omega_2{}^2 \mathrm{dt} + \int_{t_3}^{t_4} \omega_3{}^2 \mathrm{dt} \right) \right]}} - L \right) \quad \text{(Eq. 17)}$$
In this test, the calculated R by Eq. 17 will be compared with measured one. To
perform these tests, about 100 mg bulk sediments were scattered into 30 ml 0.5%
ammonia and, after that, particles larger than 20 μm particles were removed by mesh.
In this test, we should obtain suspensions with nearly monospecific coccolith. To
achieve it, in the test with *F. profunda*, coccoliths larger than 3 μm were removed by
the sinking method described in Zhang et al. (2018) and coccoliths larger than 5 μm
were removed by the same method in the test with small *Gephyrocapsa*. Briefly, the
suspension was (1) set in a 100 ml Reagent bottle sinking freely for a few hours, and
then (2) pumped out the upper 2cm. Repeat these two steps for 5–8 times until
coccoliths were purified. The sinking duration was 2 hours for *F. profunda* sample and
1.25 hours for small *Gephyrocapsa* sample, respectively.





Then 50 ml tubes with 45 ml coccolith suspensions were mounted in the centrifuge
and run with the settings shown in Table 1. After centrifugation, the upper 30 ml
supernatant was pumped out by pipette and then filtered onto 0.4 μm polycarbonate
member with a vacuum pump. The coccoliths on polycarbonate membrane were
resuspended into 20 ml diluted ammonia again and coccoliths number in the suspension
was measured with the same method described in our previous work (Zhang et al.,
2018). Finally, the separation ratio, R, was calculated by the coccolith number in the
upper 30 ml suspension divided by the total coccolith number. All the centrifuging
experiments were carried out in laboratory with temperature controlled around 20 (±1) °C
to avoid the variation of physical properties, especially the viscosity, with temperature.
**Table 1.** The settings of two tests: the density and viscosity of glycerol in 20°C, data from
Dorsey (1940); the parameters of centrifuge employed in this study: Fp and G60 represent the
experiment carried out with *F. profunda* in 70% glycerol and small *Gephyrocapsa* (<3 μ m) in
60% glycerol, respectively; L represents the minima rotation radius of centrifugation, which
represents the distance between the shaft and top of suspension as illustrated in Figure 1; A, B and
C are the terms on the left side of equal mark in Eq. 8–10.

|  | glycerol (%) | η (mPa s) | ρ (g cm⁻³) | τ | Centrifuge | L (cm) | A (s⁻¹) | B (s⁻¹) | C (s⁻¹) |
|---|---|---|---|---|---|---|---|---|---|
| **Fp** | 70% | 22.5 | 1.16 | 0.040 | TDL40B | 6.2 | $1.060\times10^6$ | $9.867\times10^4\times t$ | $1.937\times10^6$ |
| **G60** | 60% | 10.8 | 1.14 | 0.084 | DL-5B | 8.37 | $7.457\times10^5$ | $9.867\times10^4\times t$ | $2.193\times10^6$ |

**3.2 Result of experiments**

In the test, 30 ml suspension was pumped out from 45 ml suspension leading to
the initial R should be 60%. However, the intercept of calculated R is smaller than 60%
as the gravity settling in Zhang et al. (2018), because the time in the x-axis of Figure 2
is the period in which angular velocity remains constant. In other words, even the time
is set as zero, the centrifuge will still do the acceleration and deceleration processes and
coccolith will move toward the bottom. The results of observed R (dots in **Figure 2**)
are close to the theoretical values (dash lines in **Figure 2**), though a few measured


results are lower than prediction. We suggested that this difference may be caused by
coccolith loss during harvesting of the coccolith from glycerol solutions into ammonia
solution.
So far, we have obtained the coccolith movement equation in the centrifugation
and prove its correctness. In the next section, a case of coccolith separation by
centrifuging method will be carried out giving an example of separation.

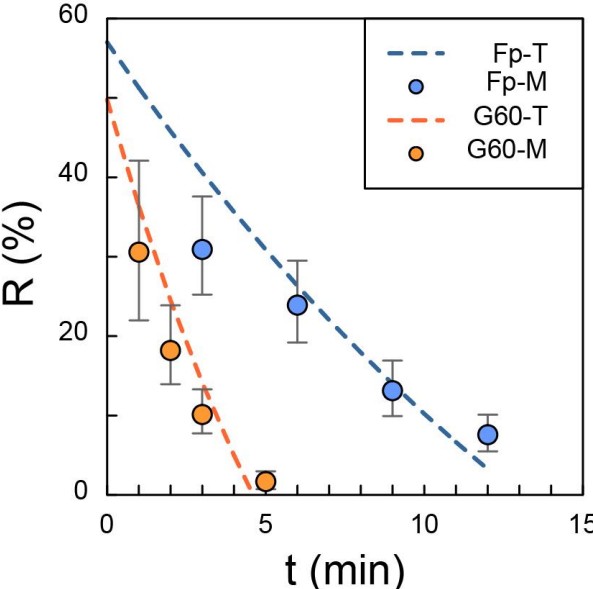


**Figure 2.** The comparison of theoretical and measured separation ratio (R): the dots
represent the measured values and dish lines are theoretical calculations. The error bars represent
95% error based on the assumption that the error of counting coccolith follows the Poisson
distribution. The orange dots represent the measured R in small *Gephyrocapsa* with 60% glycerol
test (G60-M) and the blue ones represent the measured R in *F. profunda* with 70% glycerol test
(Fp-M). The orange dashed line is the theoretical values for small *Gephyrocapsa* with 60%
glycerol test (G60-T) and the blue one is the theoretical values for *F. profunda* with 70% glycerol
test (Fp-T). The raw pictures for coccolith counting were shown in **Figure S1** and **S2**.
**4. Separation of coccoliths in practice**



### 4.1 Separation steps


The aim of this section is using the centrifugation method to separate a sample in
practice. A sample form ODP 982B (56X Section 5 5-9cm) dated around mid-Miocene
(nannofossil zoon NN4) was selected in this test. The coccolithophore *Reticulofenestra*
spp. dominated in the assemblage, with long axis length ranging from 2 μm to more
than 12 μm, offering an ideal sample to test the coccolith separation method.
*Calcidiscus* spp. (4–10 μm), *Helicosphaera* spp. (5–10 μm) and *Coccolithus* spp. (6–8
μm) were also found in this sample, which contributed less than 10% of all coccoliths
together. The preservation of fossil was moderate with many coccolith fragments but
no evidence of dissolution in the raw sample. The detailed operations are as following:
**Step 1:** weigh about 40 mg bulk sediment, scatter with 45ml 0.5% ammonia
solution and transfer the suspension into a 50 ml centrifuging tube;
**Step 2:** Calculate the centrifugation parameters (angular velocity and duration).
Here we did not measure coccolith sinking velocities, but employ the length-velocity
relationship in the previous study directly: sinking rate at $25°C = 0.0982 \times length^2$ (Zhang
et al., 2018). Based on this length-velocity equation and the centrifuge properties listed
in **Table 1**, we estimated that the angular velocity and duration for separating coccolith
with a length of 2 μm, 3 μm, 5μm, 8 μm and 10 μm should be 1850 rpm 2 min, 2250
rpm 1 min, 1400 rpm 1min, 1000 rpm 1min and 600 rpm 1min, respectively. The
Matlab code for calculating the angular velocity at fixed centrifugation duration (1 or 2
minutes) are in the supplementary.
**Step 3:** Mount the tube into the centrifuge and balance weight, set the angular
velocity as 1850 rpm and the duration as 2 minutes and start the machine;
**Step 4:** Pump out the upper 30 ml suspensions and remove them into a beaker (500
ml or larger beaker, depends how many times repeating this step) and drop about 100
μl onto a glass cover. Dry the suspension on glass cover and mount the cover on slider.
The details in this step follow Bordiga et al. (2015);
**Step 5:** Repeat Step 2–5 with different centrifugation parameters listed in Table 2;





**Step 6:** Take pictures of coccoliths in each slider on microscope and measure the
coccolith size on computer with the method described by Fuertes et al. (2014).

**Table 2**. Centrifugation parameters in the Miocene coccolith separations

|                          | <2 μm | 2–3 μm | 3–5 μm | 5–8 μm | 8–10μm |
|--------------------------|-------|--------|--------|--------|--------|
| Angular velocity ($\omega_2$, rpm) | 1850 | 2250 | 1400 | 1000 | 600 |
| Duration ($t = t_3\text{-}t_2$, s) | 120 | 60 | 60 | 60 | 60 |

**4.2 Coccolith length in each fraction**

The coccolith size distribution harvested from different centrifugation settings are

shown in **Figure 3** (the coccolith size was measured in circular polarizing microscope
and coccoliths under cross polarizing microscope were shown in **Figure S3-S9** for
species identification). The results show that the separated coccolith size increased with
the decrease of angular velocity and the differences of mean coccolith lengths are
significant between each size fractions. However, we should also notice that there is
still overlap of coccolith sizes between two neighbouring fractions. With the
centrifugation parameters set as 2250 rpm and 2 min, the coccoliths harvested have
long axis lengths around 2–4 μm and when the centrifugation parameters was varied to
1400 rpm and 1 min, the coccolith long axis size ranges from 3 μm to 7 μm, which
means coccoliths with a length between 3–4 μm appear in two fractions. Such situations
may also happen in both settling and micro filtering methods, but the range of overlap
seems to be larger for the centrifugation method compared with the size fractions
harvested by other methods.





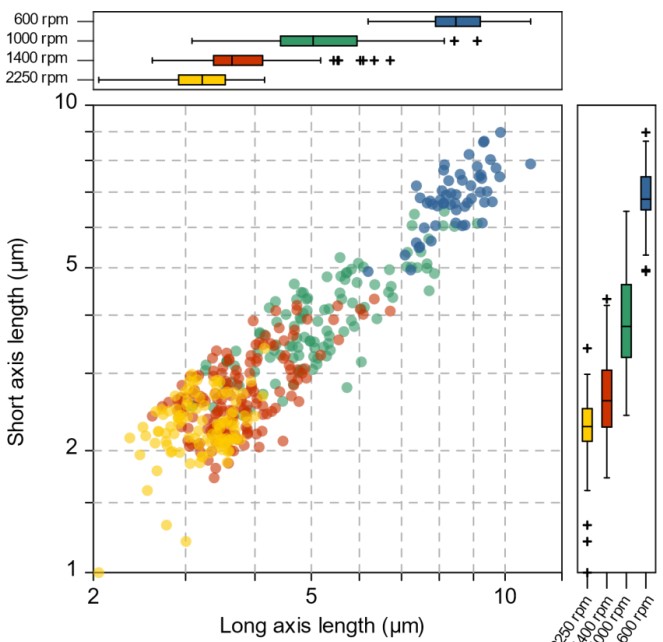

**Figure 3.** The coccolith size in different fraction after centrifuging separation: the yellow, red, green and blue dots represent 2250 rpm-2min, 1400 rpm-1min, 1000 rpm-1min and 600 rpm-1min, respectively.

### 4.3 Troubleshooting

The first potential reason leading to overlap may be the repeating times are not enough. This could be the main problem for settling under gravity, since the time costs for separation under gravity is much larger than the centrifugation method. Bolton et al. (2012) suggested that 4–6 times separations are enough for fossil extraction and in our separations, we repeated more than 8 times for a certain centrifugation setting. Considering these facts, we suggest that this overlapping was not caused by the separation times.

Another reason could be that larger coccoliths, which are supposed to sink into the lower suspension, are pumped out after centrifugation. When the upper suspension was



pumped out, the pumping speed could be too fast drawing up larger coccolith from the
lower suspension. This problem could be solved by reducing the pumping speed.
Hoverer, in practice, the pumping speed of pipette is difficult to control. Here we
recommend to modify the tips of pipette as following steps: (1) suck a drop of glue into
the top of pipette tips (the Norland optical adhesive 74 was employed in this study); (2)
solidify the glue with ultraviolet ray to seal the top of tips; (3) dill holes above the glue
horizontally. After this modification, the suspension will go into tips horizontally
instead of vertically (**Figure 4a**) to avoid mixing larger coccoliths with smaller ones.

The size overlapping could also be caused by the centrifugation tube not remaining

perfectly horizontal during centrifugation. In our calculations, the tubes are assumed to
be perfectly horizontal during all centrifugation processes and, thereby it was assumed
that there should be no collisions between coccoliths and tube wall nor among
coccoliths. However, in practice, the tubes in centrifuge are not always horizontal and
even a few degrees slope of the tubes can lead some coccoliths to knock and stick on
the tube wall forming a significant coccolith layer on one side of tube wall as illustrated
in **Figure 4b**. These coccoliths on tube wall will be pumped out after centrifugation
causing the coccolith length overlapping among two fractions. To avoid this problem,
before the step of pumping out suspension, we should observe the tube carefully. If a
coccolith layer can be found on the tube wall, the pipette tip should be placed on the
opposite of the coccolith layer to reduce the size overlapping.

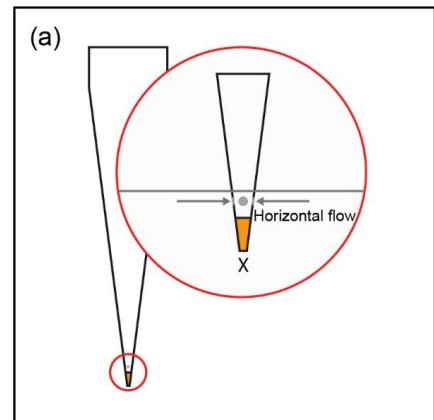 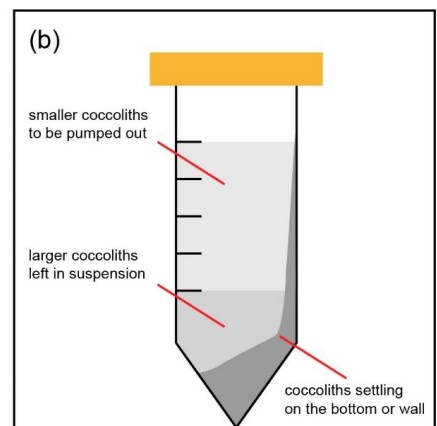





**Figure 4.** Two methods to reduce the coccolith size overlapping. (a) Adaption of pipette tip:
the orange part on tip represents sealed by solidified glue and the gray parts mean that small holes
should be drilled allowing the suspension flowing in horizontally; (b) Choose a property pumping
position to avoid extracting the coccolith on tube wall: the lightest gray part in the tube represents
the suspension in which the smaller coccolith floats, most of the larger coccoliths are in the lower

part of the suspension and the tube bottom.

**5. Summary**

In this study, we described the method of separating coccolith from bulk sediment

by centrifuge. The rotation speed for separating coccoliths within a certain range of
length could be solved after measuring the rotations radium (property of centrifuge)
and fixing the centrifugation duration.

The centrifugation method is not perfect accurate and could still mix different

species of coccolith as other traditional separating methods. The size overlapping of
this method could be reduced by adapting the pipette tips and avoiding pumping the
coccolith on tube well out. However, this method is more efficient in separating the
finest particle (smaller than 3 μm) out of bulk sediment, which is always the time-
consuming step in micro-filtering and sinking method. Thereby, this method can be
widely used in the sample preparation for analyses needing a large amount of material,
such as coccolith clumped isotope and radioactive carbon isotope measurement.
Moreover, the centrifugation method can be combined with other separation steps, for
example using the centrifugation method to remove the finest particles followed by
micro filtering with different size of membrane. This method could largely reduce the
time cost in sample preparation for coccolith geochemistry analyses and have the
potential for wide use in the future.

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

**Author contributions.**

This study was conceived by H.Z. and C.L. Measurements and calculations were
conducted by H.Z. H.Z., H.S. and L.M. wrote the paper.

**Acknowledgement**

This study was founded by National Science Foundation of China (41930536, to
C.L.) ETH core funding (to H.S), European Union's Horizon 2020 research and
innovation program under the Marie Sklodowska-Curie gran agreement (795053 to
L.M.M.) and Chinese Scholarship Council (CSC) scholarship to H.Z. We thank the
Integrated Ocean Drilling Program (IODP) for providing the samples. We thank Dr.
Guodong Jia for providing two centrifuges to test our work and Xinquan Zhou for
identification the Miocene nannofossils.