# Peer review of "Technical note: Accelerate coccolith size separation via"

_Biogeosciences, 2020_

## Referee Comment (RC1) · Anonymous Referee #1 · 25 Jan 2021

Zhang, H., C. Kiu, L.M. Mejia and H. Stoll, Technical note: Accelerate coccolith size separation via repeated centrifugation.

I enjoyed to read your manuscript. The technological approach in this article is basically well prepared. I think that this article should publish soon after several correction.

Line 71: You assume that coccolith shape is spherical. But, most coccolith is flat and disc shape. Disc shape should sink down slowly through water column. Why do you assume that coccolith shape is spherical ? What is the difference between sphere and disc shape.

Supporting Information Line 4: Figure S1. F. profunda should be italic.

General comments: How do you estimate density of sediment particles / unit volume

in water column ? High particle density should interfere each other.

Why don't you use flow cytometry method for separating small perticles in water ?

Except for these comments and questions, this is an interesting paper.

––––––––––––––––––––––––––––

---

## Referee Comment (RC2) · Anonymous Referee #2 · 26 Jan 2021

The article present new data, and the authors revise there a manuscript, figures, and Tables. I would suggest accepted to publication.

---

## Author Comment (AC1) · 26 Jan 2021

We thank for your comments and questions. Here are our responeses.

Line 71: You assume that coccolith shape is spherical. But, most coccolith is flat and disc shape. Disc shape should sink down slowly through water column. Why do you assume that coccolith shape is spherical? What is the difference between sphere and disc shape.

Response: Here we did not assume that coccoliths are spherical. We calculate the sinking process of spherical particles just to prove that the assumption that particles reach termination sinking almost at once. The sinking speed of coccolith is about 29% as that of spherical particles with same size (Zhang et al., 2018, Figure 6a). We will

make it clear in our manuscript.

Supporting Information Line 4: Figure S1. F. profunda should be italic.

Response: Done.

How do you estimate density of sediment particles / unit volume? High particle density should interfere each other.

Response: This is a very good question. Since it is difficult to estimate the number of particles in the suspension, we use the bulk sediment mass in a certain amount of to liquid to replace the real density of particles. We reconmand that no more than 400mg bulk sediment in 100ml suspension to aviod a significant sinking velocity reduction (∼5%). We estimate this number in the previous work (Zhang et al., 2018) following the work by Richardson and Zaki (1954) and assuming that the sediment is composed of 50% calcite (with a density of 2.7 g cm-3) and 50% clay (about 1.7 g cm-3).

Why don't you use flow cytometry method for separating small particles in water?

Response: There are two works on coccolith/coccolithophore separation based on flow cytometry (Halloran et al., 2009; Langley et al., 2020), which provide another fast and convenience solution. However, for many laboratories, a flow cytometry is not always affordable compared with centrifuge. Our method does not need specific equipment which could benefit for most of groups willing to work on coccolith isotope records.

Reference:

Halloran, Paul R., Nigel Rust, and Rosalind EM Rickaby. "Isolating coccoliths from sediment for geochemical analysis." Geochemistry, geophysics, geosystems 10.3 (2009). Langley, Beth, et al. "A new method for isolating and analysing coccospheres within sediment." Scientific reports 10.1 (2020): 1-13. Richardson, J. F., and W. N. Zaki. "The sedimentation of a suspension of uniform spheres under conditions of viscous flow." Chemical Engineering Science 3.2 (1954): 65-73. Zhang, Hongrui, et al. "A refinement of coccolith separation methods: measuring the sinking characteristics of

coccoliths." Biogeosciences 15.15 (2018): 4759-4775.

---

## Author Comment (AC2) · 26 Jan 2021

Thanks for your revision.
* * *